# Impulse Oscillometry as a Diagnostic Test for Pulmonary Emphysema in a Clinical Setting

**DOI:** 10.3390/jcm12041547

**Published:** 2023-02-15

**Authors:** Allan Klitgaard, Anders Løkke, Ole Hilberg

**Affiliations:** 1Department of Internal Medicine, Lillebaelt Hospital, Beriderbakken 4, 7100 Vejle, Denmark; 2Department of Regional Health Research, University of Southern Denmark, J.B. Winsløws Vej 19, 3, 5000 Odense, Denmark

**Keywords:** pulmonary emphysema, impulse oscillometry, body plethysmography, diagnostic accuracy, pulmonary function test, computed tomography, chronic obstructive pulmonary disease

## Abstract

Body plethysmography (BP) is the standard pulmonary function test (PFT) in pulmonary emphysema diagnosis, but not all patients can cooperate to this procedure. An alternative PFT, impulse oscillometry (IOS), has not been investigated in emphysema diagnosis. We investigated the diagnostic accuracy of IOS in the diagnosis of emphysema. Eighty-eight patients from the pulmonary outpatient clinic at Lillebaelt Hospital, Vejle, Denmark, were included in this cross-sectional study. A BP and an IOS were performed in all patients. Computed tomography scan verified presence of emphysema in 20 patients. The diagnostic accuracy of BP and IOS for emphysema was evaluated with two multivariable logistic regression models: Model 1 (BP variables) and Model 2 (IOS variables). Model 1 had a cross-validated area under the ROC curve (CV-AUC) = 0.892 (95% CI: 0.654–0.943), a positive predictive value (PPV) = 59.3%, and a negative predictive value (NPV) = 95.0%. Model 2 had a CV-AUC = 0.839 (95% CI: 0.688–0.931), a PPV = 55.2%, and an NPV = 93.7%. We found no statistically significant difference between the AUC of the two models. IOS is quick and easy to perform, and it can be used as a reliable rule-out method for emphysema.

## 1. Introduction

Chronic obstructive pulmonary disease (COPD) is a disease characterized by airflow limitation and respiratory symptoms usually caused by exposure to noxious gases and particles [1]. It is increasingly recognized as a complex and heterogenous disease, and patient phenotypes within COPD has been studied using various approaches [2,3], and currently accepted clinical phenotypes include the eosinophilic phenotype and the emphysema phenotype [4]. Emphysema is characterized by destruction of lung parenchyma, resulting in an abnormal increase in space distal to the terminal bronchioles [5]. The phenotyping of emphysematous COPD is usually based on computed tomography (CT), which is currently the most precise procedure [6,7]. The distinction is important, because patients with emphysema have worse prognosis [8,9], different characteristics [10,11,12], and other potential treatment options compared to patients with non-emphysematous COPD [13].

The role of pulmonary function tests (PFT) in diagnosing and monitoring the development of pulmonary emphysema is of great interest, as these tests hold the potential to reduce the need for CT scans. The most widely used and tested PFT is spirometry [14], and the ratio of forced expiratory volume in the first second (FEV1) over forced vital capacity (FVC) is an essential criterion in the diagnosis of COPD. A combination of symptoms and spirometry has recently been shown to identify COPD patients with high likelihood of having emphysema [15]. Body plethysmography (BP) provide additional characteristics such as lung residual volume (RV) and total lung capacity (TLC). RV, TLC, and the ratio of RV/TLC can be used to determine the degree of hyperinflation [16], which is indicative of pulmonary emphysema.

Impulse oscillometry (IOS) is a PFT that measures airway resistance and reactance by oscillating sound waves of different frequencies through the respiratory system [17,18]. The resistance at 5 Hz (R5) represents the total airway resistance, and the resistance at 20 Hz (R20) represents the resistance of the proximal/large airways. The resistance of the distal/small airways is therefore R5 minus R20 (R5-20). X5 is the reactance at 5 Hz, and the frequency response (Fres) is the frequency at which the total reactance is zero. The reactance area (Ax) is the total reactance at all frequencies from X5 to Fres. IOS is performed during normal tidal breathing and requires minimal patient cooperation. In contrast, spirometry can be challenging for elderly and patients with cognitive impairments, poor motor skills, and breathing difficulties [19,20,21,22,23,24]. 

IOS has been described in COPD [25,26,27,28,29], but no previous studies have investigated the diagnostic value of IOS in emphysema. We evaluated the diagnostic accuracy of IOS for emphysema, compared to BP, in a pulmonary outpatient clinic.

## 2. Materials and Methods

### 2.1. Study Design

One hundred consecutive patients with pulmonary symptoms of all causes were included in a cross-sectional study to investigate the additional value of IOS in the investigation of pulmonary symptoms at the pulmonary outpatient clinic at Lillebaelt Hospital, Vejle, Denmark, between October 2018 and March 2019. A CT scan was performed in 88 of the patients. These 88 patients were included in the present study, which specifically investigates the diagnostic accuracy of IOS in emphysema. 

### 2.2. Data Collection

Data were collected retrospectively via patient records. A clinical assessment was performed in all patients on the day of inclusion, including disease history, a BP with hemoglobin-adjusted diffusing capacity for carbon monoxide (DLCOc % of predicted), and an IOS. BP was performed with the Jaeger MasterScreen Body, and IOS was performed with the Vyntus IOS, both combined with the SentrySuite software. Variables for symptoms were collected as binary variables (yes/no) according to presence or absence of the symptom. Comorbidities were collected in the same way, with one or more diseases within a defined domain resulting in a “yes”. Smoking history was defined as never smoker, former smoker, and current smoker. For this study, we have computed a binary variable of smoking history, where former and current smoker was combined and defined as having a positive smoking history.

### 2.3. Computed Tomography Scans

CT scans were performed either in relation to the present investigation or in relation to investigation of pulmonary symptoms prior to inclusion. Scans performed up to 3 years prior to study inclusion were included and evaluated by both a radiologist and an experienced pulmonologist (OH). They were visually scored for emphysema severity on a scale from 0 to 3 (0: No emphysema, 1: Mild emphysema, 2: Moderate emphysema, 3: Severe emphysema). For this study, we have computed a binary variable for the presence or absence of emphysema.

### 2.4. Statistical Analysis

The population was stratified based on the presence of emphysema on the CT scan. The two groups were compared using two-sample *t*-test or Wilcoxon rank-sum test for continuous variables and Fischer’s exact test for categorical variables. 

We considered a selected range of variables for multivariable logistic regression analyses based on relevance for the diagnosis of emphysema. The aim was to obtain simple, comparable, and generalizable models. Clinical variables were age, sex, body mass index (BMI), dyspnea, cough, and smoking history. BP variables were FEV1 % of predicted, FEV1/FVC, RV, TLC, and DLCOc % of predicted. IOS variables were R5 % of predicted, R5-20, Fres, X5, and Ax. The diagnostic accuracy of each of these variables was evaluated individually, with area under the ROC curve (AUC) for continuous variables and sensitivity, specificity, positive predictive value (PPV) and negative predictive value (NPV) for categorical variables. For continuous variables, where higher values were associated with a lower probability of emphysema, the values were inverted.

We then constructed two logistic regression models. Model 1 consisted of clinical variables and BP variables. Model 2 consisted of clinical variables and IOS variables. We included all considered variables and performed a stepwise backwards elimination, using likelihood ratio testing for nested models and Akaike Information Criterion (AIC) for non-nested model comparison. All clinical variables could be eliminated without affecting the models, except for smoking history in Model 2, and we included smoking history in both models.

The remaining variables showed high multicollinearity, and we eliminated multicollinear variables based on AIC and link test for model specification. Overall goodness of fit was evaluated with Stukel’s test. In Model 1, we included the combined variable of RV/TLC, instead of RV and TLC individually. FEV1 % of predicted and FEV1/FVC were highly collinear, and we included only FEV1/FVC in the final model. In Model 2, R5 % of predicted and R5-20 were highly collinear. We included only R5 % of predicted based on lower AIC, better model specification, and a significantly higher AUC. Ax and X5 were highly collinear, and only Ax was included in the final model. 

We computed sensitivity, specificity, PPV, and NPV for both models, based on the cut-off point where sensitivity and specificity intersected. The AUCs were compared using the DeLong test for comparison of ROC curves. We then performed 5-fold cross-validated AUCs (CV-AUC). 

To explore potential bias in the exclusion of the 12 patients without a CT scan, we evaluated the difference in PFT parameters between the excluded patients and the included patients (Table A1 in Appendix A). Only DLCOc % of predicted showed a statistically significant difference, and the excluded group had a DLCOc % of predicted comparable to the non-emphysematous group. We also computed logistic regression models for all 100 patients under the assumption that the 12 excluded patients did not have emphysema. These models were not significantly different from the models with 88 patients (Table A2 and Figure A1 in Appendix A).

STATA 17 was used for all statistical analyses.

## 3. Results

Patient characteristics stratified by presence of emphysema can be seen in Table 1. In total, 20 patients had emphysema (9 had mild emphysema, 8 had moderate emphysema, and 3 had severe emphysema). There were no differences in sex, age, BMI, cough, and comorbidities between the two groups. Sixteen patients (80%) had dyspnea in the group with emphysema, compared to 35 patients (51%) in the group without emphysema. Sixteen patients (80%) had a positive smoking history in the group with emphysema, compared to 26 patients (38%) in the group without emphysema. Although the patients included had different causes for their respiratory symptoms, there were no significant differences with regards to CT scan results between the two groups. All variables from BP, apart from FVC, were significantly different between the two groups, and all variables from IOS, apart from R5% of predicted, were significantly different between the two groups.

The diagnostic accuracy of all the individual variables considered for the regression models is displayed in Table 2. Age and BMI had AUCs of 0.539 (95% CI: 0.408–0.670) and 0.547 (95% CI: 0.399–0.696), respectively. The PPVs of all categorical variables were <40%. The NPVs of having female sex, dyspnea, cough, and a positive smoking history were 75% (95% CI: 59–87), 89% (95% CI: 75–97), 79% (95% CI: 67–89), 91% (95% CI: 78–98), respectively.

The AUC for the BP parameters, FEV1 % of predicted, FEV1/FVC, RV/TLC, and DLCOc % of predicted, were 0.842 (95% CI: 0.728–0.955), 0.874 (95% CI: 0.766–0.981), 0.743 (95% CI: 0.621–0.865), and 0.795 (95% CI: 0.682–0.908), respectively. The AUC for the IOS parameters, R5 % of predicted, R5-20, X5, Fres, and Ax, were 0.592 (95% CI: 0.431–0.754), 0.718 (95% CI: 0.568–0.867), 0.630 (95% CI: 0.462–0.798), 0.742 (95% CI: 0.603–0.881), and 0.704 (95% CI: 0.537–0.870), respectively.

Table 3 shows the results of the two logistic regression models. Smoking history was a significant predictor of emphysema in Model 2 (IOS), but not in Model 1 (BP). In Model 1, the variables FEV1/FVC and DLCOc % of predicted were significant predictors of emphysema, with odds ratios (OR) of 0.86 (95% CI: 0.79–0.94) and 0.95 (95% CI: 0.90–0.995), respectively. RV/TLC was not a significant predictor in this model. Diagnostic accuracy of Model 1: AUC = 0.905 (95% CI: 0.805–1.00), CV-AUC = 0.892 (95% CI: 0.654–0.943), sensitivity = 84.2%, specificity = 83.8%, PPV = 59.3%, and NPV = 95.0%. In Model 2, the variables R5 % of predicted and Ax were significant predictors of emphysema, with ORs of 0.97 (95% CI: 0.94–0.995) and 9.12 (95% CI: 1.76–47.36), respectively. Fres was not a significant predictor in this model. Diagnostic accuracy of Model 2: AUC = 0.861 (95% CI: 0.767–0.955), CV-AUC = 0.839 (95% CI: 0.688–0.931), sensitivity = 80.0%, specificity = 80.9%, PPV = 55.2%, and NPV = 93.7%. When R5-20 was included instead of R5 % of predicted, the AUC changed from 0.861 to 0.783, which was a significant decrease (*p* = 0.037).

The ROC curves for the regression models are shown in Figure 1 Model 1 had a slightly larger AUC, but the difference between the two models was not significant (*p* = 0.37).

## 4. Discussion

We have evaluated the diagnostic accuracy of IOS for pulmonary emphysema in a population of patients with respiratory symptoms of all causes. IOS was compared to BP, including oxygen diffusing capacity, and it showed an excellent discriminatory ability (AUC > 0.80) [30]. Although BP had a higher AUC than IOS, we did not find the difference to be statistically significant.

IOS has mainly been investigated in asthma, especially in children, because of its minimal requirement for patient cooperation [31,32]. However, IOS has also been investigated in COPD as an alternative to spirometry, and the interest in IOS is expected to increase further [31,33]. Several studies have shown correlation between IOS parameters and both spirometry parameters and COPD severity [25,28,34,35], and IOS may even detect early manifestations of COPD better than spirometry can [29,36]. However, IOS is not yet recommended as a PFT in the diagnosis of COPD for several reasons, e.g., the lack of accepted reference values and evaluation under different disease conditions [37].

A key feature of IOS is its effectiveness in the detection of small airways disease (SAD), which is highly related to emphysema [38], and it may be more effective in this aspect than spirometry [31,37]. A study by Su et al. found IOS to be superior to spirometry in detecting SAD in COPD, where especially Fres and R5-20 were of importance [39], and IOS has been shown to discriminate better than spirometry between central and peripheral obstruction in emphysema [40]. However, Crim et al. found only a modest correlation between IOS parameters and severity of emphysema on CT in the largest study on IOS and COPD to date, with 2054 patients from the ECLPISE study [28].

The inclusion of R5 % of predicted in Model 2 resulted in a significantly better AUC than when including R5-20. This contrasts with the univariate ROC curve analyses seen in Table 2, where R5-20 is one of only two predictors with AUC > 0.7. In previous studies examining IOS in COPD, R5-20 was more strongly correlated with spirometry obstruction parameters than was R5Hz [25,34], and a recent study has also found R5-20 to be correlated with emphysema on CT [41]. Our results indicate that the diagnostic accuracy of IOS is more dependent than BP on clinical parameters, such as age, sex, and BMI. These parameters are already adjusted for in the R5 % of predicted variable, why the R5 % of predicted variable may be more important than R5-20 in the multivariable analysis that excludes the clinical variables. This may also explain why smoking history is significant only in Model 2.

Spirometry and BP have also been investigated in relation to emphysema. A systematic review from 2012 investigated the correlation between CT emphysema measurements and airway obstruction parameters in spirometry [42]. Both FEV1 % of predicted and FEV1/FVC were significantly correlated with CT emphysema measurements, with FEV1/FVC showing the strongest association. This agrees with results from our present study (Table 2). Several studies have shown that TLC measured by BP correlates with the degree of emphysema on CT [43,44,45,46]. The correlation between emphysema, including emphysema severity, and RV/TLC is also well established [47,48]. Our current study confirms this relationship; RV, TLC, and RV/TLC were significant univariate predictors of emphysema (Table 1). However, our data suggest that RV/TLC is not a significant predictor of pulmonary emphysema when the diffusing capacity and the ordinary spirometry variable FEV1/FVC are also included, and this also agrees with existing literature. Studies have shown a stronger association between emphysema and diffusion capacity than between emphysema and lung volume parameters, including RV/TLC [47,48,49]. Furthermore, Kahnert et al. found that BP may be redundant in the evaluation of emphysema, when compared to the combination of spirometry and diffusion capacity [11], and our results further acknowledges this understanding.

In our study population, 9 patients had mild emphysema, 8 had moderate emphysema, and 3 had severe emphysema. It is difficult to determine how this distribution could affect our results, as the literature on IOS in emphysematous copd is extremely limited. As IOS may be better in detecting SAD than spirometry, and SAD is linked to emphysema with SAD likely preceding emphysematous destruction of lung tissue [38], it may be speculated that IOS is better at detecting mild and early emphysema. On the other hand, the BP parameters RV, TLC, and DLCOc are likely affected only when emphysema has progressed to a certain degree, and BP might be advantageous in detecting moderate-to-severe emphysema. This is, however, only speculation.

There are some limitations to this study. The small study sample increases the risk of type 2 errors, and the study population is heterogenous. The study includes 88 patients with pulmonary symptoms of different causes, and the results may not be directly applicable in a population of patients with COPD. However, this study also has several strengths. Because of the small study sample, we created diagnostic models that are small and simple, which enhances replicability and generalizability. This has the added value of being easily understandable and applicable by clinicians in the daily setting. Another strength of the study is the cross validation of the AUCs, which further enhances generalizability.

## 5. Conclusions

In conclusion, IOS showed excellent diagnostic accuracy for emphysema. BP showed a higher AUC than did IOS in the discrimination between presence and absence of emphysema, but the difference was not statistically significant. IOS is both quick and easy to perform, and it can be used when BP is not possible and as a reliable rule-out method for emphysema in clinical settings where time and space is limited. IOS may be a helpful alternative tool in the detection and management of emphysematous COPD, and studies investigating these qualities in a selected population of patients with COPD may elevate the usefulness of IOS in COPD.

## Figures and Tables

**Figure 1 jcm-12-01547-f001:**
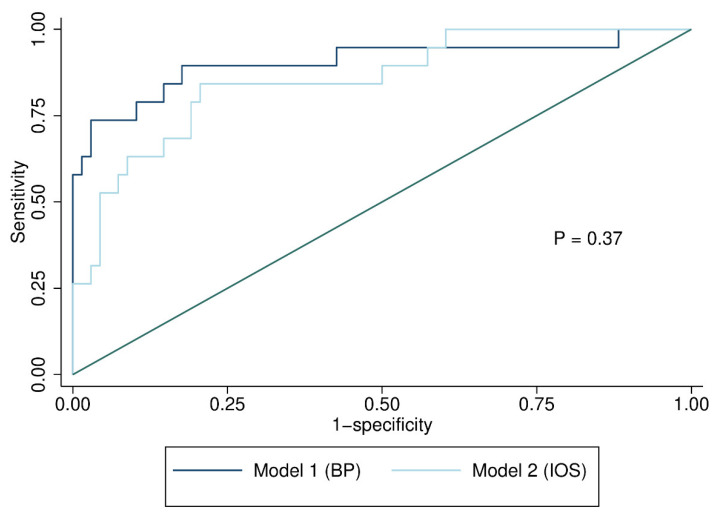
ROC curves for the two logistic regression models of body plethysmography (BP) and impulse oscillometry (IOS) variables.

**Table 1 jcm-12-01547-t001:** Patient characteristics stratified on presence of emphysema.

	Emphysema (n = 20)	No Emphysema (n = 68)	*p* *
**Clinical variables**	**n (%)**	**n (%)**	
Female sex	9 (45)	36 (53)	0.62
Positive smoking history	16 (80)	26 (38)	<0.01
Dyspnea	16 (80)	35 (51)	0.04
Cough	8 (40)	18 (26)	0.27
Comorbidities			
- Cardiovascular	7 (35)	27 (40)	0.80
- Neurological	2 (4)	3 (10)	0.32
- Musculoskeletal	3 (15)	11 (16)	1.00
- Endocrinological	4 (20)	7 (10)	0.26
- Gastroenterological	3 (15)	8 (12)	0.71
- Psychiatric	5 (7)	2 (10)	0.66
- Active cancer	2 (3)	1 (5)	0.37
CT scan results			
- Bronchiectasis	10 (50)	19 (28)	0.10
- Airtrapping	11 (55)	20 (29)	0.06
- Sarcoidosis	0 (0)	11 (16)	0.06
- Infection	1 (5)	6 (9)	1.00
- Fibrosis	5 (25)	14 (21)	0.76
	**Median (IQR)**	**Median (IQR)**	
Age (years)	63 (52–72)	62 (47–72)	0.60
BMI	26.6 (23.3–31.2)	26.3 (23.3–29.7)	0.50
**Body Plethysmography**	**Median (IQR)**	**Median (IQR)**	
FEV1	1.7 (0.9–2.6)	2.8 (2.2–3.2)	<0.01
FEV1 % of predicted	67 (43–83)	97 (83–107)	<0.01
FVC	3.1 (2.2–4.1)	3.6 (3.2–4.2)	0.07
FVC % of predicted	99 (77–112)	104 (95–112)	0.02
FEV1/FVC	52.7 (46.2–63.3)	75.3 (70.2–80.4)	<0.01
TLC	7.0 (5.3–7.8)	5.8 (5.2–6.8)	0.04
TLC % of predicted	109 (100–132)	97 (88–107)	<0.01
RV	3.5 (2.6–4.7)	2.4 (2.0–3.0)	<0.01
RV % of predicted	153 (129–217)	114 (93–131)	<0.01
RV/TLC	0.52 (0.44–0.64)	0.40 (0.34–0.48)	<0.01
DLCOc % of predicted	56 (43–63)	73 (65–82)	<0.01
**Impulse Oscillometry**	**Median (IQR)**	**Median (IQR)**	
R5 % of predicted	108 (80–141)	98 (77–120)	0.06
R5-20	0.14 (0.06–0.25)	0.06 (0.03–0.10)	<0.01
X5	−0.15 (−0.28–−0.09)	−0.10 (−0.13–−0.07)	<0.01
Fres	21.17 (15.46–24.16)	14.08 (10.50–18.16)	<0.01
Ax	1.18 (0.29–2.90)	0.32 (0.16–0.81)	<0.01

* Calculated with two-sample t-test or Wilcoxon Rank-sum test for continuous variables as appropriate. Fischer’s exact test was used for categorical variables. Abbreviations: IQR, inter quartile range; BMI, body mass index; FEV1, forced expiratory volume in the first second; FVC, forced vital capacity; TLC, total lung capacity; RV, residual volume; DLCOc, hemoglobin-adjusted diffusing capacity for carbon monoxide; R5, resistance at 5 Hz; R5-R20, resistance at 5 Hz minus resistance at 20 Hz; X5, reactance at 5 Hz; Fres, frequency response; Ax, reactance area.

**Table 2 jcm-12-01547-t002:** Diagnostic accuracy of each individual variable considered for logistic regression compared to reference standard outcome *.

	Sensitivity, % (95% CI)	PPV, % (95% CI)	Specificity, % (95% CI)	NPV, % (95% CI)	AUC (95% CI)
**Clinical variables**					
Age	-	-	-	-	0.539 (0.408–0.670)
BMI	-	-	-	-	0.547 (0.399–0.696)
Female sex	45 (23–69)	20 (10–35)	47 (35–60)	74 (59–87)	-
Dyspnea	80 (56–94)	31 (19–46)	49 (36–61)	89 (75–97)	-
Cough	40 (19–64)	31 (14–52)	74 (61–84)	81 (69–90)	-
Positive smoking history	80 (56–94)	38 (24–54)	61 (48–72)	91 (78–98)	-
**Body Plethysmography**					
FEV1 % of predicted **	-	-	-	-	0.842 (0.728–0.955)
FEV1/FVC **	-	-	-	-	0.874 (0.766–0.981)
RV/TLC	-	-	-	-	0.743 (0.621–0.865)
DLCOc % of predicted **	-	-	-	-	0.795 (0.682–0.908)
**Impulse Oscillometry**					
R5 % of predicted	-	-	-	-	0.592 (0.431–0.754)
R5-20	-	-	-	-	0.718 (0.568–0.867)
X5 **	-	-	-	-	0.630 (0.462–0.798)
Fres	-	-	-	-	0.742 (0.603–0.881)
Ax	-	-	-	-	0.704 (0.537–0.870)

* Reference standard outcome: Emphysema present on CT scan. ** Values inverted. Abbreviations: PPV, positive predictive value; NPV, negative predictive value; AUC, area under the ROC curve; CI, confidence interval; BMI, body mass index; FEV1, forced expiratory volume in the first second; FVC, forced vital capacity; TLC, total lung capacity; RV, residual volume; DLCOc, hemoglobin-adjusted diffusing capacity for carbon monoxide; R5Hz, resistance at 5 Hz; R5-R20Hz, resistance at 5Hz minus resistance at 20 Hz; X5Hz, reactance at 5 Hz; Fres, frequency response; Ax, reactance area.

**Table 3 jcm-12-01547-t003:** The two multivariate logistic regression models to discriminate between presence and absence of emphysema on CT scan.

	Model 1 (BP)		Model 2 (IOS)	
	OR (95% CI)	*p*	OR (95% CI)	*p*
Positive smoking history	1.45 (0.28–7.45)	0.66	4.39 (1.11–17.39)	0.04
FEV1/FVC	0.86 (0.79–0.94)	<0.01	-	-
RV/TLC	5.80 (0.002–20,480.37)	0.42	-	-
DLCOc % of predicted	0.95 (0.90–0.995)	0.04	-	-
R5 % of predicted	-	-	0.97 (0.94–0.995)	0.03
Fres	-	-	1.00 (0.81–1.25)	0.97
Ax	-	-	9.12 (1.76–47.36)	<0.01
**Diagnostic accuracy (95% CI)**				
AUC	0.905 (0.805–1.00)		0.861 (0.767–0.955)	
CV-AUC	0.892 (0.654–0.943 *)		0.839 (0.688–0.931 *)	
Sensitivity, %	84.2 (50.0–94.1 *)		80.0 (61.9–94.7 *)	
PPV, %	59.3 (35.5–77.8 *)		55.2 (40.0–73.1 *)	
Specificity, %	83.8 (64.3–91.5 *)		80.9 (53.6–93.2 *)	
NPV, %	95.0 (85.7–98.2 *)		93.7 (87.5–97.7 *)	

* Bootstrap bias corrected CI. Abbreviations: OR, odds ratio; AUC, area under the ROC curve; CV-AUC, k-fold cross-validated AUC; PPV, positive predictive value; NPV, negative predictive value; CI, confidence interval; FEV1, forced expiratory volume in the first second; FVC, forced vital capacity; TLC, total lung capacity; RV, residual volume; DLCOc, hemoglobin-adjusted diffusing capacity for carbon monoxide; R5Hz, resistance at 5 Hz; R5-R20Hz, resistance at 5Hz minus resistance at 20 Hz; X5Hz, reactance at 5 Hz; Fres, frequency response; Ax, reactance area.

## Data Availability

The data presented in this study are available upon request. The data are not publicly available due to restrictions related to the study approval as this was a quality assurance project.

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
