# Peer review of "Impulse Oscillometry as a Diagnostic Test for Pulmonary Emphysema in a Clinical Setting"

_jcm, 2023, doi:10.3390/jcm12041547_

Round 1
Reviewer 1 Report
Introduction: It's ok
Methods:
- More information about the equipment used por BP and IOS
- Population: About
- About CT scans: How many radiologists interpreted the CTs? Was there more than one radiologist who interpreted the same CT? How was the emphysema's score created or what is it based on? Is it possible to quantify emphysema from CT scans?
Results: Could you add more data on the patients in Table 1, such as other comorbidities, heart disease, pulmonary hypertension?
Reviewer 2 Report
As this is a journal for clinicians, I would like to suggest to the authors begin by explaining what COPD is and its phenotypes and then go to the reasons for the need to differentiate the emphysematous phenotype.
Page 3, line 113: Please clarify what is "AIC".
In the Discussion section, the Authors expose the two main fragilities of the study, i.e. the small study sample and the inclusion of patients with pulmonary symptoms of different causes. This should be stated clearly in the methods, and in the results, and better addressed in the discussion. The idea when reading the paper until this point is that all individuals were COPD patients, either with emphysematous or non-emphysematous phenotypes, discriminated by CT scans. The other diagnosis should be stated in the results.
Would the authors please comment on the choice of treating the emphysema severity as a binary variable (other than the small number of patients, if there is one) and expose in the results (table 1) the numbers in mild, moderate, and severe categories and address in discussion how this distribution could affect the results and conclusions?
Round 2
Reviewer 1 Report
No further comments.
Reviewer 2 Report
The authors satisfactorily answered the questions.